# Multiparametric Ultrasound of Cervical Lymph Node Metastases in Head and Neck Cancer for Planning Non-Surgical Therapy

**DOI:** 10.3390/diagnostics12081842

**Published:** 2022-07-30

**Authors:** Julian Künzel, Moritz Brandenstein, Florian Zeman, Luisa Symeou, Natascha Platz Batista da Silva, Ernst Michael Jung

**Affiliations:** 1Department of Otorhinolaryngology, Head and Neck Surgery, University Hospital of Regensburg, 93053 Regensburg, Germany; luisa.symeou@ukr.de; 2Department of Radiology, University Hospital of Regensburg, 93053 Regensburg, Germany; moritz.brandenstein@stud.uni-regensburg.de (M.B.); natascha.platz-batista-da-silva@ukr.de (N.P.B.d.S.); ernst-michael.jung@ukr.de (E.M.J.); 3Center of Clinical Studies, University Hospital of Regensburg, 93053 Regensburg, Germany; florian.zeman@ukr.de

**Keywords:** CEUS, shear wave elastography, perfusion, neck metastases, superficial lymph nodes

## Abstract

Background: We aimed to evaluate multiparametric ultrasound, to achieve a better understanding of the baseline characteristics of suspected cervical lymph node metastases in head and neck cancer before induction chemotherapy or chemoradiation. Methods: From February 2020 to April 2021, our complete ultrasound examination protocol was carried out on clinically evident malignant lymph nodes of histologically proven HNSCC in the pre-therapeutic setting. Results: A total of 13 patients were eligible for analysis. Using elastography, irregular clear hardening in areas in the center of the lymph node could be detected in all cases. Elastographic Q-analysis showed a significantly softer cortex compared to the center and surrounding tissue. The time–intensity curve analysis showed high values for the area under the curve and a short time-to-peak (fast wash-in) in all cases compared to the surrounding tissue. A parametric evaluation of contrast enhanced the ultrasound in the early arterial phase and showed an irregular enhancement from the margin in almost all investigated lymph nodes. These results show that the implementation of comprehensive, multiparametric ultrasound is suitable for classifying suspected lymph node metastasis more precisely than conventional ultrasound alone in the pre-therapeutic setting of HNSCC. Thus, these parameters may be used for improvements in the re-staging after chemoradiation or neoadjuvant therapy monitoring, respectively.

## 1. Introduction

In head and neck squamous cell cancer (HNSCC), approximately 50% of patients present at advanced stages with evident lymph node (LN) metastases. For a long time, only B-scan and color-coded duplex imaging (CCDI) have been the mainstay in ultrasound (US) diagnostics of cervical LNs. In recent years, several innovative US features such as THI (tissue harmonic imaging) and Crossbeam have been developed and implemented in modern US machines. Moreover, power Doppler and B-Flow are available for a better visualization of vascular LN architecture. Although US allows one to evaluate not only the size of an LN, but also to look for the cystic configuration, hyperechoic lesions, the loss of fatty hilum, extracapsular extension and peripheral vascularization, which are considered to be the major criteria for LN malignancy, its specificity is not satisfying [1,2]. Except for size, these values are hardly comparable as they are not objective criteria. Based on the assumption that tissue with a malignant transformation shows increased rigidity, this provides a complementary diagnostic parameter that is largely independent of the examiner’s subjective assessment [3]. At the current time, the elastography of cervical LNs is to be regarded as a useful additional procedure, but needs further standardization in order to improve the reproducibility and validity [4]. Several methods and scores are currently utilized to evaluate the stiffness of LNs, such as real-time elastography, strain elastography with semi-quantitative Q-analysis, and shear wave elastography [5]. In combination with other established methods, elastography could increase the accuracy in the differential diagnosis of cervical LN [6,7]. The shear wave elastography is much more independent of the pre-compression and, therefore, more reproducible and has been well established in the staging of liver fibrosis and of breast lesions, possibly also in thyroid gland lesions, where its use increases the sensitivity and specificity of US examinations [5,8,9]. Currently, the use of US elastography for the measurement of therapy response is mainly limited to gastroenterology, but the first results of anticancer therapy monitoring have been published for various indications [10,11]. Bhatia et al. obtained the best result in the differential diagnosis of LN by using the mean elasticity modulus at a cut-off value of 30.2 kPa, where the sensitivity and specificity were 41.9% and 100%, respectively [12]. Choi et al. published a cut-off using the maximum elasticity modulus by 19.4 kPa, with a sensitivity of 91% and a specificity of 97% [13]. Others showed that elastography in combination with a B-scan achieves values of 92% sensitivity and 94% specificity in the diagnosis of malignant LNs in the neck area [14]. Later on, other authors found significant cut-offs for shear wave elastography of 3.34 m/s and 3.27 m/s with a sensitivity of 78.9–88.6% and a specificity of 74.4–94.1% in the distinction between benign and malignant cervical LNs [15] [16]. A meta-analysis from 2017 showed a sensitivity of 81% and specificity of 85% for shear wave elastography in the diagnosis of cervical LNs [17]. Furthermore, US elastography may be beneficial for identifying the most suspicious areas within an LN to be targeted for sampling [4].

Changes in the intranodal vascular architecture, such as displacement, an aberrant course, an avascular focus and subcapsular vessels, which suggest malignant degeneration, was reported by Tschammler et al. more than 20 years ago [18]. We know that the presence of metastatic foci in a LN alters the internal blood flow and that US allows such changes to be detected at an early stage [19]. The detailed display of the vascular anatomy can be facilitated by adding contrast agents to CCDI. Moreover, for the detection of the capillary perfusion pattern, it is really necessary to use the contrast-enhanced ultrasound (CEUS) by reduced mechanical index (MI; <0.2) [20,21]. Second generation contrast agents, such as sulfohexafluoride microbubbles, enable, by oscillation, a dynamic evaluation of the micro vascularization up to capillary perfusion with a sensitivity and specificity of 93% and 88%, respectively [1,21]. In the diagnosis of abdominal LNs, CEUS has already proven itself to be a reliable method [22]. Furthermore the combination of conventional US and CEUS had the highest diagnostic accuracy (92.7%) compared with conventional US (80.8%) and CEUS (89.1%) alone in the diagnosis of LN metastasis from papillary thyroid carcinoma [23]. CEUS is already an established method in the therapy monitoring of liver tumors, as well as for other relevant indications, and offers an additional pathophysiological insight during and after a therapy sequence through the possibility of a high-resolution investigation of the perfusion kinetics, whereby the time–intensity curve (TIC) perfusion analysis represents a quantitatively measurable variable [24,25,26,27]. Time–intensity curve perfusion analysis has so far only been investigated a little in the assessment of cervical LN metastases and CEUS, for the characterization of superficial lymphadenopathy, at present, has not been recommended in guidelines for routine clinical use [21]. Modern software-supported analysis tools help us today to determine objectifiable measured values and make, for example, the parametric and time–intensity curve perfusion analysis of CEUS possible. A meta-analysis by Mei showed a potentially high value of CEUS in the diagnosis of superficial LN, but the authors call for a standardization of the examination protocols in order to further establish the method in clinical routine [28]. In our working group, a corresponding test protocol has already been successfully used for various indication areas and the results have been published accordingly [2,29,30,31]. The working group also has experience in the use of CEUS in cervical LN metastases [1], showing a strong tendency towards a shortened time-to-peak and an increased contrast enhancement (AUC = area under the curve) in malignant LNs. The state of the art assessment of the therapy response after induction chemotherapy or chemoradiation is performed by PET/CT [32,33], but the metabolic response may be inconclusive [34] and a routine PET-based follow-up may not be achievable for all head and neck cancer patients as PET resources may be limited. In this regard, the advantages of US diagnostics are the quick implementation, low costs, lack of radiation exposure and the possibility of direct interventional biopsy. Our own investigations were aimed to retrospectively evaluate routinely acquired general parameters of B-scans, fundamental CCDI, strain elastography, shear wave elastography and parametric considerations, including the time–intensity curve analysis of CEUS, in order to achieve a better understanding of the values in the interdisciplinary pre-therapeutic baseline assessment of suspected cervical LN metastases in HNSCC before the start of induction chemotherapy or chemoradiation. The data of multiparametric analysis could contribute to a further development of artificial intelligence in the evaluation. The incorporation of a comprehensive pre-therapeutic US work-up of LN metastases in the routine staging algorithms at our Head and Neck Cancer Center may open up the possibility for US guided therapy monitoring during neoadjuvant therapy or for a tailored follow-up after chemoradiation [22,35]. Here, we present the first results and describe our examination protocol.

## 2. Materials and Methods

We use and combine the technical innovations of US, the high-resolution CEUS, with parametric perfusion analysis as well as shear wave elastography in the interdisciplinary pre-therapeutic staging of advanced HNSCC (cN+), intended for chemoradiation and/or induction chemotherapy, in addition to routine diagnostics such as contrast-enhanced CT or MRI of the neck. Before the USexamination was performed, written informed consent was signed by all patients undergoing “off label” CEUS examination. Since February 2020, our complete routine US examination protocol (Table 1) was carried out on most patients with histologically proven advanced HNSCC with clinically evident LN metastases. All patients were planned for either induction chemotherapy or definitive chemoradiation according to tumor board consensus. A combination of established standard B-Scan and CCDI criteria were used to classify clinically suspected LNs as metastases (short axis diameter > 8 mm; Solbiati-Index < 2, hypo- or hyperechoic lesions, loss of fatty hilum, extracapsular extension and peripheral vascularization). A single center retrospective multiparametric analysis of eligible routine cases examined between February 2020 and April 2021 was carried out.

The local institutional review board granted a waiver in accordance with the Declaration of Helsinki because of the retrospective nature of the performed multiparametric analysis of routinely acquired image data (Ethics Committee of the University of Regensburg; 20-1949-104).

The routine examination protocol of each patient with histologically proven HNSCC started by identifying clinically suspected LN metastases of the neck by B-Scan. Unenhanced color coded and power Doppler US as well as B-Flow were performed to evaluate the vascular anatomy and spreading of vessels within the LNs. The LN was measured in two planes and characteristic patterns such as shape, boarders, hilum sign and vascularity have been described. These measures were carried out by an experienced Head and Neck Surgeon (JK) and the findings have been confirmed by an experienced Radiologist (EMJ), each with high resolution multifrequency linear probes (ML 6–15 MHz and L6–9 MHz, LOGIQ E9, GE Healthcare GmbH, Solingen, North Rhine-Westphalia, Germany). The elastography and CEUS were all carried out by EMJ using the same probes. In performing elastography, the node beneath the probe was deformed by a “push pulse” generated from the probe. Strain elastography was coded using a five scale false colors code (red, orange, yellow, green/turquoise, blue). Then the velocity of the shear waves propagating within the tissue was detected, and the stiffness was assessed based on the detected shear velocity in the largest possible diameter of the LN.

Thereafter, CEUS was performed after intravenous bolus injection of 2.4 mL sulfohexafloride microbubbles (SonoVue^®^, BRACCO Imaging, Milan, Italy), flushed with 10 mL of a physiological saline solution. Furthermore, LN perfusion and enhancement dynamics were analyzed. During the examination, the US probe was placed on the patients’ skin and fixed in a stable position focused on the preassigned LN. CEUS was started from the time of the bolus injection of the microbubbles. EMJ stored the DICOM cine loops and pictures of elastography and CEUS perfusion from the LN over more than 1 min with parallel imaging of CCDI and fundamental B-scan, the true agent detection mode of CEUS. This mode of parallel imaging of B-scan and CEUS makes it easier to detect pathological findings. For the routine work-up of our patients, EMJ wrote standardized reports including at least a qualitative description of the stiffness in strain elastography and one measure for shear wave elastography as well as a qualitative description of wash-in and wash-out kinetics of CEUS.

For the implementation of the retrospective study protocol (Table 1), an independent experienced reader (MB) used the DICOM raw data. All analyses were performed with the machine integrated workstation (LOGIQ E9, GE Healthcare GmbH, Solingen, North Rhine-Westphalia, Germany). The stiffness of the LN was repeatedly (3×) measured by shear waves in kPa and m/s using a ROI drawn with the largest possible diameter not extending beyond the LN margins. The values of the repeated measures were used to build a mean value for overall LN stiffness in kPa and m/s. Q-analysis of strain elastography with Q-ratio was performed by placing two ROI in the center, four at the margin of the LN and two in the surrounding soft tissue. The ROI within the LN included areas with the lowest and the highest stiffness. With the assumption that the stress is uniformly distributed throughout the field of view, the strain in the ROI can be compared to an ROI in the reference tissue that is experiencing a similar stress. This provides a semi-quantitative measurement of the relative rather than absolute tissue stiffness. Q-analysis reflects the stiffness change of the tissue during 2 ms. In our setting, the first ROI placed in the center of the LN was defined as reference. Evaluation of the relative stiffness (rU) of the center of the suspicious LN in correlation to the margin and the surrounding tissue was performed with values from 1 rU to a maximum of 6 rU. The diameter of each ROI was 2–3 mm. Time–intensity curve analysis is an integrated analysis software of the US device (LOGIQ E9, GE Healthcare GmbH, Solingen, North Rhine-Westphalia, Germany) for the quantification of tissue microvascularization. The reliability of this tissue quantification tool has been described in previous investigations of our group and details of time–intensity curve analysis are published in the EFSUMB CEUS non-liver guidelines [21,30,36,37]. Briefly, the average signal intensity of the contrast agent wash-in and wash-out is displayed as time–intensity curve analysis. In post-processing, several regions of interest (ROI) can be set and the perfusion characteristics of the different ROIs can be extracted from the time–intensity curve. The perfusion parameter time-to-peak correlates with the velocity of microvascularization and is given in seconds (s), whereas area under the curve correlates with the blood volume in the evaluated tissue area and is given in relative units (rU). We placed again two ROI in the center, four at the margin and two in the surrounding soft tissue (outside of vessels). For the correct placement of ROI, only positive values were accepted (adjustment for artefacts). The simultaneous analysis of more than eight ROIs was not possible. The diameter of each ROI was 2–3 mm. Using the integrated software of the US machine, the time–intensity curve analysis started at the early arterial phase after 10–15 s (first microbubbles visible) up to 1 min, recorded with DICOM cine loops. In addition to that, parametric evaluation of the perfusion kinetics was performed. The first 10 s of enhancement by the LN in the early arterial phase were analyzed using an integrated software tool (LOGIQ E9, GE Healthcare GmbH, Solingen, North Rhine-Westphalia, Germany). A false color code was used to evaluate micro vascularization. Red shows early and high enhancement, followed by orange, yellow, green, blue and deep blue for less vascularization.

The statistic tests were performed by an independent statistician (FZ). Results are presented using median (1st quartile, 3rd quartile); minimum–maximum for all parameters. Differences between center, cortex and surrounding tissue were analyzed by using Friedman′s test with unadjusted post-hoc pairwise comparisons in case of a significant *p*-value. A *p*-value < 0.05 was considered as statistically significant for all statistical tests. Analyses were performed using SPSS version 26.0 (IBM SPSS, version 26.0, Armonk, New York, NY, USA).

## 3. Results

Our complete examination protocol could be applied in sixteen patients, but only thirteen cases (two women and eleven men) where eligible for retrospective analysis (three induction chemotherapy, ten chemoradiation). The patients’ age ranged from 49 to 72 years (mean 60.15). The review of the three ineligible cases showed that in one case the selected LN did not meet clear B-Scan criteria for malignancy and was, therefore, excluded from analysis. In the two other cases, movement artefacts of tracheotomized patients hindered optimal time-intensity curve analysis.

With B-scan morphology, a clinically suspected malignant LN could be localized in all 13 patients with histologically proven HNSCC. In all cases with parameters adapted to low flow (PRF/scale < 1000 Hz, wall filter < 100 Hz, color gain highly adapted) for CCDI, power Doppler and especially for B-Flow (see Appendix A), it could be demonstrated that metastatic LNs showed an irregular vascularization pattern. The short axis diameter was over 8 mm (mean: 1.84 cm; range: 0.86–4.28 cm; SD: 0.92 cm) for all selected LN metastases (Figure 1).

With the elastography in the strain technique, irregular clear hardening in areas of the center of the LN could be detected in all cases (Figure 2).

The mean velocity of all 13 LNs was 4.17 m/s, corresponding to 55.83 kPa. The absolute and relative Q-analysis showed a significant softer cortex compared to the center and the surrounding tissue (Figure 3a,b; Table 2). The elasticity of the center of the LN did statistically not differ from the surrounding tissue.

At the end of CEUS perfusion analysis there is a single image including measurement values for dynamic CEUS parameters such as time-to-peak and area-under-the-curve perfusion parameters (Figure 4) [21]. In this single image analysis of the CEUS (cine loops up to 90 sec), the typical contrast medium behavior of the clinically suspected metastatic LNs in all cases was a rapid and irregular enhancement from the cortex to the hilar medulla. In all cases, after the strong contrast enhancement up to the end of the arterial phase, there was an incipient wash-out and thus a curve-drop of up to 90 s. However, a residual enhancement could still be present. The time–intensity curve analysis showed high values for the area under the curve and a short time-to-peak in all cases compared to the surrounding tissue (Table 2 and Figure 4).

As shown in Table 2, the surrounding tissue showed significantly lower values for the area under the curve than the cortex and center. Furthermore, the time-to-peak was significantly shorter in the center compared to the surrounding tissue, both corresponding to the hyper- and neovascularization of metastases.

The parametric evaluation of CEUS by false colors in the early arterial phase (up to 8 s) showed an early irregular contrast enhancement from the margin (red) in almost all investigated LNs (Figure 5a). This was followed by the transition of microbubbles to the center and the hilum coded orange and yellow. Thereafter, the contrast reached the surrounding tissue (Figure 5b).

The visualization of necrotic non-vascularized areas within the LN could be clearly visualized by parametric evaluation (Figure 6). The complete implementation of the time–intensity curve analysis of our study protocol on the US machine took 10–15 min. However, this also included a movement correction for each individual ROI.

## 4. Discussion

For head and neck oncologists, high-resolution US is an extremely important procedure, alongside the clinical examination, for the accurate staging of HNSCC. Although US is an examiner-dependent modality, the findings are also reproducible, if they are measured by using standardized protocols, including DICOM cine loops and stored in digital archiving systems. The resolution of LNs on US is better than that of CT and MRI. Ultrasound is also cheaper and without radiation exposure. In our study, clinically suspected LN metastases could be clearly detected by using internationally accepted B-scan and CCDI criteria, recently summarized by Künzel et al. [2]. Even though we did not take tissue from the examined LNs, the examined LNs showed clear signs of malignancy, as published extensively in the literature, and all patients had histologically diagnosed HNSCC. We are aware of the fact that the gold standard of the histological confirmation of malignancy was missing in the investigated LNs, but this represents the real clinical setting in patients planed for neoadjuvant therapy or simultaneous chemoradiation. The very limited number of examined patients is another obvious limitation of the presented data, but the selected indication for this protocol did not allow the acquisition of more cases within 14 months in this monocentric setting. Due to the retrospective nature of this study, it was not possible to obtain the same post-therapeutic parameters in a standardized manner, whereof these data were not suitable for analysis. The majority of the included patients received chemoradiation at external radiotherapeutic departments in south-east and north-east Bavaria and, therefore, it was not possible to guarantee a standardized multiparametric US protocol for monitoring the treatment response in this retrospective study setting. The main limitations for multiparametric US work-up in general are that the examinations themselves and the time–intensity curve analysis are time consuming, require an experienced examiner, a special high resolution linear probe for CEUS and a special software for perfusion analysis. Especially in HNSCC cases, it may not be possible for the patients to lie quietly on the examination bed long enough, which causes movement artefacts, comparable to MRI or PET/CT. Moreover, often only one LN can be examined with one bolus of contrast agent, monitoring its dispensation over a certain period. Therefore, the examiner needs to apply the standard criteria for malignant LN metastases to select the most suspicious LN as the target for monitoring. In this regard, it is mandatory to clearly describe the position of the target lesion by landmarks and store the images and cine loops accordingly. However, standardized US monitoring of LNs on-demand without repeated radiation exposure or expensive MRI studies may be an innovative approach to a safe and tailored oncologic surveillance of advanced HNSCC.

Our own data show a mean shear wave elasticity of 55.83 kPa (4.17 m/s) in the clinically suspected LN metastases, which is a clear sign of hardness and stiffness in these nodes [3,4,12,13,14,15,16,17]. These values are well comparable with published cut-offs [12,15,16]. In the elastographic Q-analysis we could demonstrate that the absolute and relative values of the cortex of the LN were significantly softer than the center of the LN and the surrounding tissue of the neck (Table 2). The elasticity of the center of the LN did not statistically differ from the surrounding tissue. These conditions should change if the metastases are responding to therapy. Although several publications have already shown only a marginal improvement in the differential diagnosis of benign vs malignant LN, we think that accurately analyzing elastographic measures including Q-ratio analysis can help in the follow-up and therapy monitoring of neck metastases. However, shear wave techniques make it possible to carry out quantitative measurements in real time with US, and they might, in the future, be used to validate standard and threshold values, as in echocardiography [15]. A problem to be solved, in this way, is the wide variety of different soft- and hardware solutions, which need to be standardized.

Using US contrast media in the multiparametric assessment of malignant superficial LNs makes it possible to generate quantitative data, which could be used for therapy monitoring [18,19,22,23]. However, this procedure is, at present, not yet standard in routine clinical work, as the measurement parameters still vary widely [1,25,38]. Ye and colleagues have already shown that the peak intensity and time-to-peak in the early phase (after the 5th irradiation dose) of radiation therapy predict a treatment response of LN metastases from nasopharyngeal cancer with high sensitivity and specificity. The time-to-peak increased from 10.98 to 12.30 s, with the peak intensity decreasing in all patients during the course of therapy (sensitivity 94.3%, specificity 88.2%) [35]. In this study, we only examined clinically suspected LN metastases in cases of histologically proven HNSCC. The results of the time–intensity curve analysis in our study showed high values for the area under the curve and a shortened time-to-peak in all pre-therapeutic LN metastases, thus allowing good parameters for follow-up. The surrounding tissue showed significantly lower values for the area under the curve than the cortex and center and a significantly shorter time-to-peak in the center compared to the surrounding tissue. This supports the hyper- and neovascularization of metastatic LN. According to other previously published results, the typically fast wash-in with a short time-to-peak is well comparable with our results [39]. By using additional parameters, such as the wash-in rate, rise time and wash-out rate, local perfusion changes of solid lesions could be detected [37]. In the parametric analysis of CEUS, we were able to demonstrate that the perfusion pattern distributes from the margin to the center of the LNs in almost all cases. Software analysis tools enable a subsequent calculation of the parametric perfusion pattern (e.g., centripetal or centrifugal), which could be additionally helpful to visualize the perfusion pattern and maybe incorporate artificial intelligence into the US staging of HNSCC [40]. Undoubtedly, CEUS helps to better assess the dignity of LNs by showing the micro- and macro-vascularization and may enable US to objectively monitor therapy response.

Our first results show that the implementation of a standardized multiparametric US protocol including elastography techniques and CEUS in the pre-therapeutic staging algorithm of advanced HNSCC is feasible. Comprehensive elastographic and perfusion analysis describe LNs more precisely than conventional US alone, because multiparametric US delivers many objectifiable values. However, detailed multiparametric analysis is time consuming and may, therefore, be reserved for selected cases which may profit, such as patients before the start of chemoradiation. Our next step is to design a prospective trial with larger sample size to correlate the pre-therapeutic with the intra- and post-therapeutic values in the same standardized manner to learn how the measures are changing. The future perspective is a clinical prospective study, ideally in a multicentric setting, comparing the standardized conventional response assessment by means of CT, MRI or PET/CT with state-of-the-art multiparametric US diagnostics.

## Figures and Tables

**Figure 1 diagnostics-12-01842-f001:**
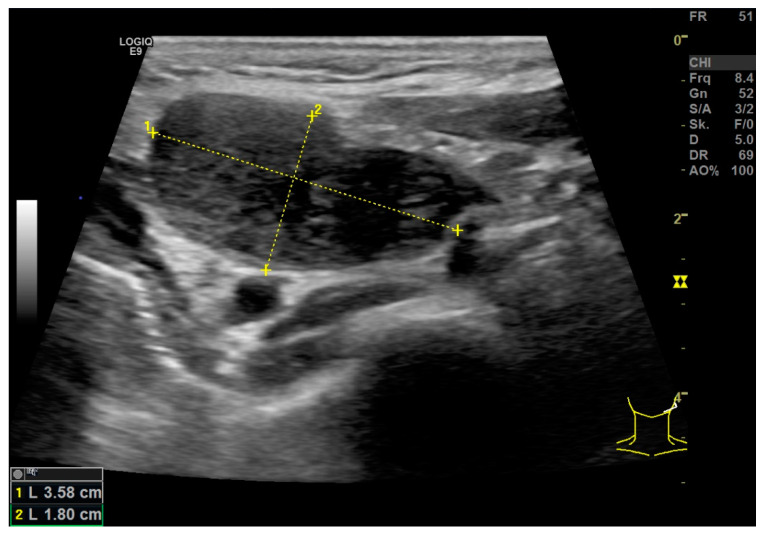
B-scan of malignant lymph node in neck level II of the left side, short axis diameter (2) of 1.80 cm, Solbiati-Index 1.98, necrotic areas, absence of hilum sign.

**Figure 2 diagnostics-12-01842-f002:**
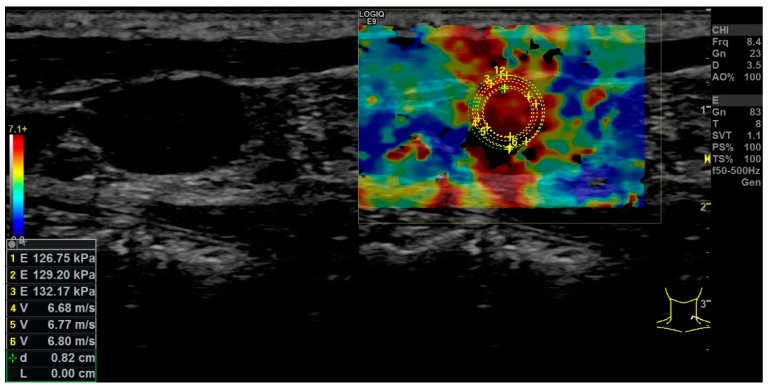
Example of shear wave elastography (right side) of a selected lymph node of the left neck. The left side of the picture shows the parallel B-scan detection mode. Values of repeatedly measured stiffness in kPa (E) and m/s (V) are shown on the left side. The stiffness of the tissue is presented by a false color code with red representing hard tissue and blue soft tissue.

**Figure 3 diagnostics-12-01842-f003:**
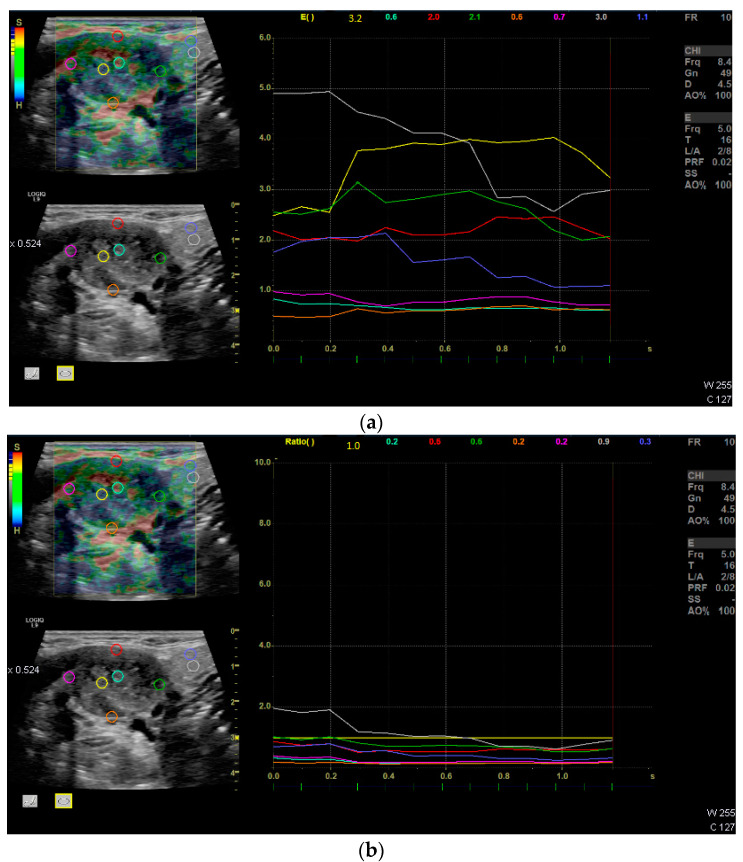
Suspected lymph node metastases of the right neck Level II. (**a**) Example of relative units (multicolored lines) of Q-analysis in strain elastography over time; Regions of interest (multicolored circles) are placed in the center and the cortex of the lymph node and in the surrounding tissue; Graph: x-axis: time in seconds (s); y-axis: relative tissue stiffness displayed as semi-quantitative measures in relative units. (**b**) Q-ratio analysis; yellow region of interest is set as reference (1.0), the other colored lines are put in relation.

**Figure 4 diagnostics-12-01842-f004:**
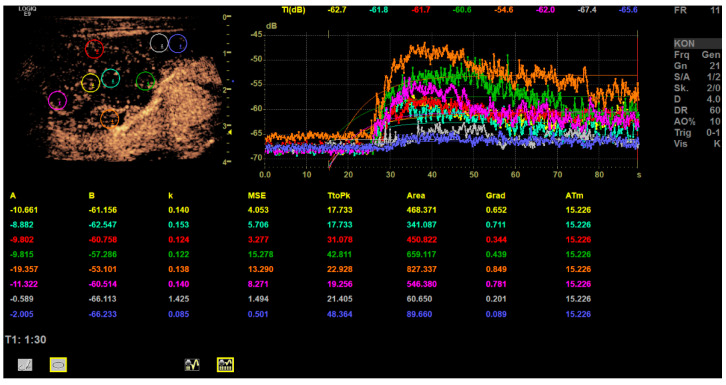
Example of a time–intensity curve analysis with 6 ROIs (multicolored circles) placed in a lymph node of the left neck level II and 2 ROIs (grey and deep blue circle) placed in the surrounding tissue. The graph on the right side shows the perfusion kinetics over time inside the ROIs, illustrated as multicolored curves. Description of parameters in the figure: Wash-in formula: Curve value at time t = F (t) = A (1 − e − kt) + B; Wash-out formula: Curve value at time t = F (t) = A (e − kt) + B [t is the time relative to the start frame that the user has picked, not relative to t = 0. So, in the fit, it is really (t-Start), where the system very much assumes that the user has set the start frame at the time of contrast arrival, so Start is the same as T arrival (time of arrival)]; B is a baseline parameter [it represents the intensity at the time of arrival]; A is the amplitude scale factor [it describes the peak at time t = infinity for wash-in formula, and t = 0 for wash-out formula. This is why you need to have an end time for wash-in that includes a portion of the curve that is flattened out. If the curve is still rising at the end of the fit, the A value will be higher than the peak at the last frame. If the data have flattened out, the data amplitude at the user-selected last frame will be approximately the same as the data amplitude at t = infinity, so the A value in the fit will accurately reflect the peak seen in the data]; k is the exponential decay factor [it tells how fast the wash-in or wash-out happened. The larger k is, the quicker the wash-in or wash-out is]; MSE is the mean standard error [how much the real data at each time t are different from the value of the fit, F (t), at the same time t. It is the “goodness of fit.” The smaller the MSE, the better the fit is]; TtoPk = time-to-peak in seconds (s); Area = area under the curve.

**Figure 5 diagnostics-12-01842-f005:**
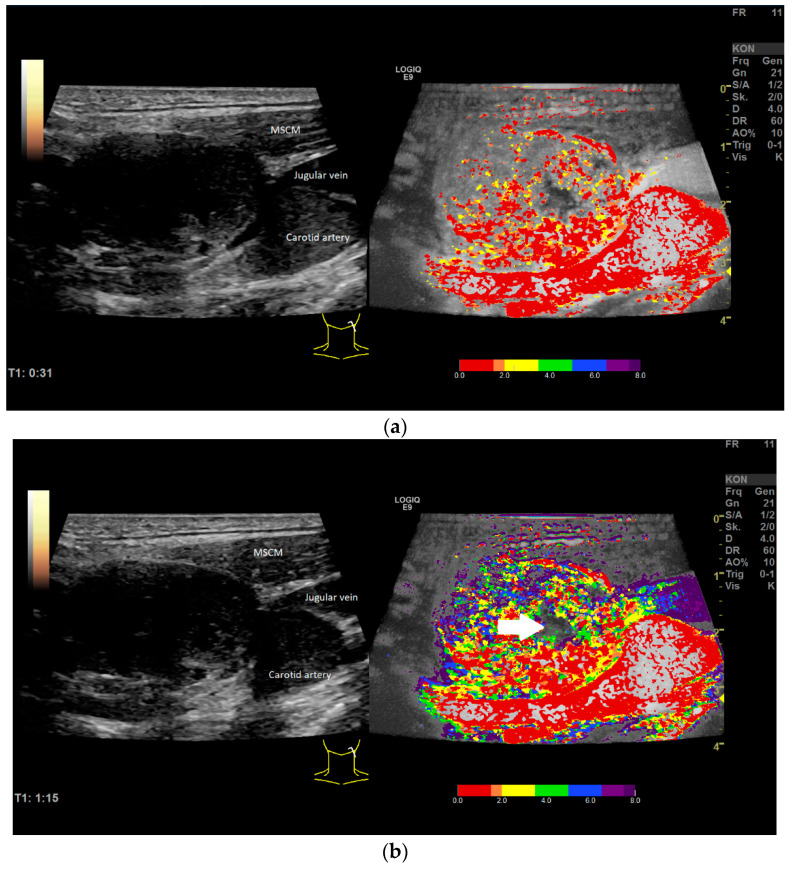
(**a**) Example of parametric evaluation of CEUS showing the contrast distribution up to approximately 2.5 s, coded by false colors (see color bar, 0–8 s) in the early arterial phase; irregular contrast enhancement from the margin (red, orange); same lymph node as shown in Figure 4. (**b**) Distribution of contrast agent within the first 8 s of the arterial phase, illustrated by false colors from red (high and early enhancement) to deep blue (less and later enhancement); white arrow showing necrosis.

**Figure 6 diagnostics-12-01842-f006:**
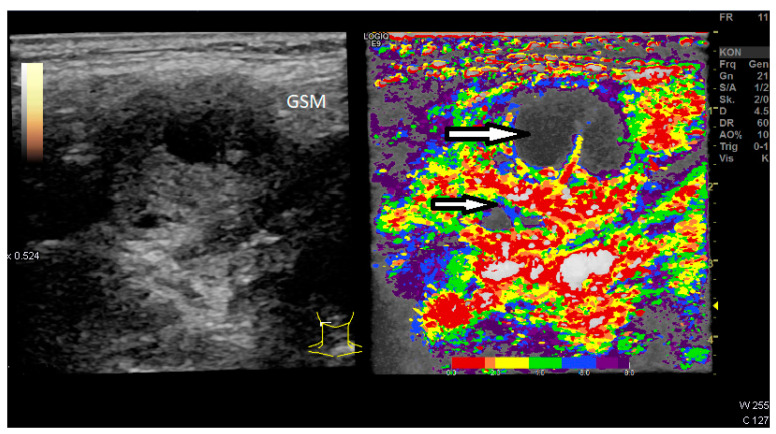
Two necrotic areas of the lymph node (Level II right neck) are clearly visualized (arrows), GSM: Submandibular gland. Distribution of contrast agent within the first 8 s of the arterial phase illustrated by false colors from red (high and early enhancement) to deep blue (less and later enhancement).

**Table 1 diagnostics-12-01842-t001:** Overview of routine multiparametric ultrasound examination protocol and additionally performed analysis according to study protocol.

Method	Routine Protocol	Study Protocol
B-Scan	Identification of clinically suspected LN metastases	-
CCDI incl. B-Flow	Vascularization pattern	-
Strain Elastography	Qualitative report of stiffness (soft or hard)	Q-analysis of strain elastography with Q-ratio; two ROI in the center, four at the margin of the LN and two in the surrounding soft tissue. The ROIs within the LN included areas with the lowest and the highest stiffness. The diameter of each ROI was 2–3 mm.
Shear Wave Elastography	Single ROI drawn with the largest possible diameter not extending beyond the LN margins (kPa).	Repeated (3x) measurement by kPa and m/s using ROI drawn with the largest possible diameter not extending beyond the LN margins. The values of the repeated measures were used to build a mean value for overall LN stiffness in kPa and m/s.
CEUS	Perfusion and enhancement dynamics (Wash-in/Wash-out)	Time–Intensity Curve analysis with Time-to-Peak and Area-under-the-Curve; two ROI in the center, four at the margin of the LN and two in the surrounding soft tissue (outside of vessels). For the correct placement of ROI, only positive values were accepted (adjustment for artefacts). The diameter of each ROI was 2–3 mm. Additionally parametric evaluation of the perfusion kinetics.

CCDI: color coded duplex imaging; CEUS: contrast enhanced ultrasound; LN: lymph node; ROI: region of interest.

**Table 2 diagnostics-12-01842-t002:** Statistics of Q-Analysis of elastography and CEUS perfusion analysis (n = 13).

	Center (1)	Cortex (2)	Surrounding Tissue (3)	Global *p*-Value ^1^	Post-Hoc Pairwise Comparisons		
					(1) vs. (2)	(1) vs. (3)	(2) vs. (3)
absolute Q-analysis in rU	2.73 (1.86, 4.18); 1.72–5.13	1.57 (1.25, 2.80); 0.94–3.83	3.70 (2.01, 4.22); 0.48–5.50	0.009	0.011	0.845	0.006
relative Q-analysis in rU	1.03 (0.80, 1.24); 0.17–1.37	0.60 (0.56, 0.85); 0.31–1.23	1.25 (0.60, 1.90); 0.10–2.75	0.023	0.031	0.695	0.011
mean values of TTP in s	21.85 (18.08, 34.47); 12.06–72.46	28.88 (20.55, 36.52); 18.14–56.02	39.19 (25.72, 50.10); 17.35–59.37	0.023	0.170	0.006	0.170
mean values of AUC in rU	499.6 (307.1, 677.2); 35.83–1059	538.7 (415.4, 776.5); 292.0–1212	164.01 (119.6, 199.5); 50.2–394.0	0.001	0.556	0.003	<0.001

AUC = area under the curve; Center: ROI 1–2; Cortex: ROI 3–6; Surrounding tissue: ROI 7–8; Data show median (q1, q3); min-max.; TTP = time-to-peak in seconds (s); rU = relative Units; ^1^ Friedman.

## Data Availability

Not applicable.

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
