# Peer review of "Multiparametric Ultrasound of Cervical Lymph Node Metastases in Head and Neck Cancer for Planning Non-Surgical Therapy"

_diagnostics, 2022, doi:10.3390/diagnostics12081842_

Round 1

Reviewer 1 Report

The authors have presented an interesting work with a potentially high clinical impact.

The power of modern ultrasound imaging technology is applied to the critical domain of diagnosing / assessing the pathologic nature of lymph nodes in HNSCC and a variety of state-of-the-art image-processing methods are applied.

While the work itself is sound the presentation needs some improvement with respect to understandability leading to a comprehensive presentation of results.

Formally it might be helpful to use tables to present a brief overview of the methods applied, especially in the Materials and Methods Section. This would help the readers fully appreciating the content.

The clinical study is purely exploratory and is designed such that only one (experienced) ENT specialist was performing the measurements. This will obviously raise some doubts about the generalizability of results and will have to be addressed when discussing the results.

on line 124 it is stated that "the stiffness of the LN was repeatedly (3x) measured by kPa ...." which is unclear and needs to be improved.

line 36: please explain the "true agent detection mode"

line 138: English: "values WERE"

line 140: What is the "special integrated software"? Explanation for this is absolutely required.

Results section:

Table 1 would benefit from a clearer presentation by putting the (ROI xx) into the legend of the table. Moreover, units are missing for the SE parameters. TIC analysis showing up in the results would need a more detailed explanation in the materials and methods section.

line 181: The legend to Fig. 2 seems insufficient to allow non-US-specialists to appreciate the content presented. A detailed description of the figure's content is mandatory.

Table 2: the presentation of results with respect to the statistical analysis would benefit from an explanation in the materials and methods section as it is not evident why a Q-analysis was performed and what can be learned therefrom. Again, the legend is insufficient here. Would it not be better to add colums labeled median (q1, q3) and min, max? Interestingly a footnote to "Friedmann" appears which is most likely out of context? If not, the legend needs to refer to this. The last column is the most interesting but it lacks a thorough discussion later on.

Fig. 3: Here the legend is not sufficient. Specifically, this figure shows an evolution of a set of lines (right panel) and circles and color coded areas (left panel). Non-expert US users will not be able to understand the full content.

line 206: What is "single image analysis of the CEUS"?

Fig. 4: Basically the comments to Fig. 3. apply with extending to the numbers given in the lower half of the figure.

Fig. 5: how was the contrast agent applied? Which "contrasted" structures are shown in the image? Blood or lymphatic vessels or tissue? The color coding has numbers to it which need to be explained.

line 243: What is an "independend evaluation of the TIC analysis ..."?

line 253: Here it is stated that the results are reproducible without explanation and grounding on facts. As a side comment, the mentioning of the  DICOM standard should be deleted as it bears no informative content in this context.

line 263: A formulation like "we believe that there was no doubt ...." is not appropriate for a scientific paper. Please provide sound evidence for any statement.

line 267: it is unclear what the authors mean by a "retrospective study". Up to this point it is understood that a prospective exploratory study was being undertaken. This whole paragraph is unclear and needs improvement:

Line 283 ff till the end of the paragraph seems to be part of the introduction.

line 311 ff ~ line 330 seems to be part of the introduction.

line 350, the sentence "Even in the COVID-pandemic ..." can be omitted without loss of information. US is a relatively cheap modality and almost ubiquitous.

line 356: English. "Warranted" should be exchanged to "are required" or similar.

Author Response

Dear Reviewers,

We thank you very much for your efforts taken into the Review of our manuscript. We appreciate all comments and really believe that answering all questions and revising the manuscript, respectively, improved the content of our submission significantly. We marked all changes in the text in the correction mode of Microsoft word.

Hopefully the manuscript now meets the high scientific standard of the “Diagnostics”.

Kind regards in behalf of all authors,

Julian Künzel

Answers to comments of Reviewer 1:

The authors have presented an interesting work with a potentially high clinical impact.

The power of modern ultrasound imaging technology is applied to the critical domain of diagnosing / assessing the pathologic nature of lymph nodes in HNSCC and a variety of state-of-the-art image-processing methods are applied.

While the work itself is sound the presentation needs some improvement with respect to understandability leading to a comprehensive presentation of results.

Formally it might be helpful to use tables to present a brief overview of the methods applied, especially in the Materials and Methods Section. This would help the readers fully appreciating the content.

Answer: Thank you for this valuable comment. We understand that it was difficult to understand our work-up, so we added a new Table 1 presenting an overview of routine multiparametric Ultrasound examination protocol and additionally performed analysis according to study protocol. Furthermore, the complete Material and Method section was revised and restructured.

The clinical study is purely exploratory and is designed such that only one (experienced) ENT specialist was performing the measurements. This will obviously raise some doubts about the generalizability of results and will have to be addressed when discussing the results.

Answer: Thank you for your valuable comments and overall positive review result. In the submitted version we mentioned: "The identification of clinically suspected LN metastases by B-Scan und CCDI examinations were carried out by an experienced Head and Neck Surgeon (JK) and the findings have been confirmed by an experienced Radiologist (EMJ), each with high resolution multifrequency linear probes (ML 6-15 MHz and L6-9 MHz, LOGIQ E9, GE Healthcare GmbH, Solingen, North Rhine-Westphalia, Germany). The elastography and CEUS were all carried out by EMJ using the same probes." Therefore, the measurements of Elastography and CEUS were performed by an experienced radiologists applying the established multiparametric protocol of our institution. This is highlighted in the Introduction: "In our working group, a corresponding test protocol has already been successfully used for various indication areas and the results have been published accordingly (2, 18–20)."

on line 124 it is stated that "the stiffness of the LN was repeatedly (3x) measured by kPa ...." which is unclear and needs to be improved.

Answer: We added to the text: "The values of the repeated measures were used to build a mean value for overall LN stiffness in kPa and m/s (Table 2)."

line 36: please explain the "true agent detection mode"

Answer: Explanation added to the text.

line 138: English: "values WERE"

Answer: The Reviewer is right. We changed the grammar.

line 140: What is the "special integrated software"? Explanation for this is absolutely required.

Answer: The Reviewer is right. We changed the text accordingly. All measurements were performed using the integrated software of the US machine (LOGIQ E9, GE Healthcare GmbH, Solingen, North Rhine-Westphalia, Germany).

Results section

Table 1 would benefit from a clearer presentation by putting the (ROI xx) into the legend of the table. Moreover, units are missing for the SE parameters. TIC analysis showing up in the results would need a more detailed explanation in the materials and methods section.

Answer: The Reviewer is right. We decided to delete former Table 1 as the most relevant information is presented in Table 2. We added clearer descriptions to the table legend. Details of the TIC analysis have been added to the legend of figure 4, describing all of the mentioned parameters. In Material and Methods we adviced that more details about the performed TIC analysis are published in the “CEUS non liver guidelines of EFSUMB”. Parameters of strain elastography Q-analysis do not have any units.

line 181: The legend to Fig. 2 seems insufficient to allow non-US-specialists to appreciate the content presented. A detailed description of the figure's content is mandatory.

Answer: The Reviewer is absolutely right. We revised the figure legend accordingly.

Table 2: the presentation of results with respect to the statistical analysis would benefit from an explanation in the materials and methods section as it is not evident why a Q-analysis was performed and what can be learned therefrom.

Answer: Thank you for the valuable comment. We added an explanation of Q-analysis and Q-ratio to the Material and Methods section.

Again, the legend is insufficient here. Would it not be better to add colums labeled median (q1, q3) and min, max? Interestingly a footnote to "Friedmann" appears which is most likely out of context? If not, the legend needs to refer to this.

Answer: The legend was revised. The Friedman test refers to the global p-value (1) .

The last column is the most interesting but it lacks a thorough discussion later on.

Answer: Thank you for this valuable comment. We revised the discussion accordingly.

Fig. 3: Here the legend is not sufficient. Specifically, this figure shows an evolution of a set of lines (right panel) and circles and color coded areas (left panel). Non-expert US users will not be able to understand the full content.

Answer: The Reviewer is right. We revised the legend of Figure 3.

line 206: What is "single image analysis of the CEUS"?

Answer: We changed the text accordingly: "At the end of CEUS perfusion analysis there is a single image including measurement values for dynamic CEUS parameters like Time-to-Peak and Area under the Curve perfusion parameters . In this single image analysis of the CEUS (cine loops up to 60 sec), the typical contrast medium behavior of the clinically suspected metastatic LNs in all cases was a rapid and irregular enhancement from the cortex to the hilar medulla."

Fig. 4: Basically the comments to Fig. 3. apply with extending to the numbers given in the lower half of the figure.

Answer: The Reviewer is right. We revised the legend of Figure 4 and uploaded an improved Figure 4.

Fig. 5: how was the contrast agent applied? Which "contrasted" structures are shown in the image? Blood or lymphatic vessels or tissue? The color coding has numbers to it which need to be explained.

Answer: The explanation of the false colors coding was explained more in detail in the figure legend. Anatomical landmarks were added to the figure for better orientation.

line 243: What is an "independend evaluation of the TIC analysis ..."?

Answer: The Reviewer is right, this was misleading. We deleted "independent" in line 243 and added following passage to the Material and Methods section: "EMJ stored the DICOM Cine loops and pictures of elastography and CEUS perfusion from the LN over 1 min. Afterwards an independent experienced reader (MB) performed the elastography measures and time intensity curve analysis using the machine integrated workstation. "

line 253: Here it is stated that the results are reproducible without explanation and grounding on facts. As a side comment, the mentioning of the  DICOM standard should be deleted as it bears no informative content in this context.

Answer: Thank you for this comment. We believe that DICOM technology is one of the most important developments to make ultrasound reproducible because it makes the storage of Cine loops possible. Another important issue to improve reproducibility is the use of standardized protocols as presented here. We changed the passage accordingly: "Although US is an examiner-dependent modality, the findings are also reproducible, if they are measured by using standardized protocols including DICOM cine loops and stored in digital archiving systems."

line 263: A formulation like "we believe that there was no doubt ...." is not appropriate for a scientific paper. Please provide sound evidence for any statement.

Answer: The Reviewer is right. We changed the statement accordingly: "Even though we did not take tissue from the examined LNs, the examined LN´s showed clear signs of malignancy as published extensively in the literature and all patients had histologically diagnosed HNSCC."

line 267: it is unclear what the authors mean by a "retrospective study". Up to this point it is understood that a prospective exploratory study was being undertaken. This whole paragraph is unclear and needs improvement:

Answer: The Reviewer is right. We appreciate the critical comment about the study design. We rephrased and revised the whole Material and Methods section to explain the difference between the routine multiparametric work-up and the retrospectively performed elastographic and perfusion analysis. We clearly indicated that the examinations were performed routinely during staging algorithms of HNSCC patients before chemoradiation. All examinations were stored via DICOM in the digital archive and could be analyzed later.

Line 283 ff till the end of the paragraph seems to be part of the introduction.

Answer: The Reviewer is right: We moved parts of the discussion to the introduction.

line 311 ff ~ line 330 seems to be part of the introduction.

Answer: The Reviewer is right: We moved parts of the discussion to the introduction.

line 350, the sentence "Even in the COVID-pandemic ..." can be omitted without loss of information. US is a relatively cheap modality and almost ubiquitous.

Answer: The Reviewer is right. We omitted the sentence.  We changed the sentence accordingly: “Our first results show that the implementation of a standardized multiparametric US protocol including elastography techniques and CEUS in the pretherapeutic staging algorithm of advanced HNSCC is feasible. Comprehensive elastographic and perfusion analysis is feasible and suitable for classifying LN more precisely than conventional US alone”.

line 356: English. "Warranted" should be exchanged to "are required" or similar.

Answer: The Reviewer is right. We changed the last paragraph of the conclusions: “Our next step is to design a prospective trial with larger sample size to correlate the pretherapeutic with the intra- and posttherapeutic values in the same standardized manner to learn how the measures are changing. The future perspective is a clinical prospective study, ideally in a multicentric setting, comparing standardized conventional response assessment by means of CT, MRI or PET/CT with state-of-the-art multiparametric US diagnostics.”

Reviewer 2 Report

 I congratulate the authors on an exciting study exploring multiparametric US on suspected LN in Patients with Head and neck squamous cell carcinoma. The many advanced analyses using both CEUS and elastography on the same LN are a strength of the study and provide some interesting findings for future research. However, I have some considerations regarding this study's method and conclusions. 

A major limitation is the missing golden standard. The LNs have no cytological or histological examinations for malignancy, so we cannot be sure if they have squamous cell carcinoma metastasis. I would therefore suggest changing the text to "suspected LN" rather than LN metastasis in the manuscript so the reader will not get confused. Further, the title needs to be changed, and I will suggest: 

"Multiparametric Ultrasound of metastasis suspected Cervical Lymph Nodes in Patients with Head and neck squamous cell carcinoma"

Further, the authors only selected LN that was found suspected of malignancy on the b-mode US. Therefore, they do not have any control cases with benign LN to compare, which is essential in diagnostic accuracy studies to assess diagnostic specificity. Due to the retrospective design, they do not reassess the lymph nodes after treatment which would have provided interesting data to compare. 

The conclusion is: "Our first results show that standardized multiparametric US including elastography techniques and CEUS perfusion analysis is feasible and suitable for classifying LN more precisely than conventional US alone."

I am not convinced about this conclusion. First, all the included lymph nodes were suspected malignant on b-mode ultrasound, and the multiparametric US did not change the diagnosis in these patients. To state that, you need to conduct a study where multiparametric US improves the diagnostic accuracy of LN where the b-mode US classified it as benign or unknown. Please clarify how your results support the conclusions that the diagnostic accuracy of malignant LN increases? 

Second, the authors report that the TIC analysis on the US machine took 10-15 minutes. Do the authors find the analysis feasible in a clinical diagnostic workup? It would not be in our setting, where we would prefer to take an FNA (2-5 minutes) to clarify malignancy instead. 

Minor comments: 

You write the study is retrospective, but the patients' sign informed consent to participate in the CEUS examination. Please explain. 

Please specify which software you used for the TIC analysis to evaluate TTP and AUC for velocity and volume of contrast enhancement?

Please use English titles for ultrachall publications.

The manuscript has many technical abbreviations making it difficult for the reader to follow. Please use the full name of TIC, TTP, PI, AUC.. so it is easier to read. 

Author Response

Dear Reviewers,

We thank you very much for your efforts taken into the Review of our manuscript. We appreciate all comments and really believe that answering all questions and revising the manuscript, respectively, improved the content of our submission significantly. We marked all changes in the text in the correction mode of Microsoft word.

Hopefully the manuscript now meets the high scientific standard of the “Diagnostics”.

Kind regards in behalf of all authors,

Julian Künzel

Answer to the comments of Reviewer 2:

 I congratulate the authors on an exciting study exploring multiparametric US on suspected LN in Patients with Head and neck squamous cell carcinoma. The many advanced analyses using both CEUS and elastography on the same LN are a strength of the study and provide some interesting findings for future research. However, I have some considerations regarding this study's method and conclusions. 

A major limitation is the missing golden standard. The LNs have no cytological or histological examinations for malignancy, so we cannot be sure if they have squamous cell carcinoma metastasis. I would therefore suggest changing the text to "suspected LN" rather than LN metastasis in the manuscript so the reader will not get confused.

Answer: The Reviewer is right. We investigated lymph nodes of patients before chemoradiation. In these cases routinely the cN classification based on clinical or imaging findings is applied according to UICC manuals. The wording "suspected" has been added in the manuscript were appropriate.

Further, the title needs to be changed, and I will suggest: 

"Multiparametric Ultrasound of metastasis suspected Cervical Lymph Nodes in Patients with Head and neck squamous cell carcinoma"

Answer: The Reviewer is right. We changed the title to: Multiparametric Ultrasound for Pretherapeutic Baseline Diagnostics of clinically suspected Cervical Lymph Node Metastases in Head and Neck Cancer planned for non-surgical Therapy. Furthermore we changed the wording to “suspected LN metastases throughout the manuscript.

Further, the authors only selected LN that was found suspected of malignancy on the b-mode US. Therefore, they do not have any control cases with benign LN to compare, which is essential in diagnostic accuracy studies to assess diagnostic specificity.

Answer: Even though we did not take tissue from the examined LNs, the examined LN´s showed clear signs of malignancy as published extensively in the literature and all patients had histologically diagnosed HNSCC. We are aware of the fact, that the gold standard of histological confirmation of malignancy was missing in the investigated LNs, but this represents the real clinical setting in patients planed for neoadjuvant therapy or simultaneous chemoradiation.

Due to the retrospective design, they do not reassess the lymph nodes after treatment which would have provided interesting data to compare.

Answer: The Reviewer is right. We added another explanation, why we were not able to apply a standardized mpUS protocol in this study setting to the discussion. " Due to the retrospective nature of this study, it was not possible to obtain the same post therapeutic parameters in a standardized manner, whereof these data were not suitable for analysis. Many of the included patients received chemoradiation at external radiotherapeutic departments in south-east Bavaria and therefore it was not possible to guarantee a standardized multiparametric US protocol for monitoring the treatment response in this retrospective study setting."

The conclusion is: "Our first results show that standardized multiparametric US including elastography techniques and CEUS perfusion analysis is feasible and suitable for classifying LN more precisely than conventional US alone."

I am not convinced about this conclusion. First, all the included lymph nodes were suspected malignant on b-mode ultrasound, and the multiparametric US did not change the diagnosis in these patients. To state that, you need to conduct a study where multiparametric US improves the diagnostic accuracy of LN where the b-mode US classified it as benign or unknown. Please clarify how your results support the conclusions that the diagnostic accuracy of malignant LN increases? 

Answer: We understand the concern of the Reviewer, but we do not state that our results improve the diagnosis of malignancy. Therefore, we rephrased the paragraph: “Our first results show that the implementation of a standardized multiparametric US protocol including elastography techniques and CEUS in the pretherapeutic staging algorithm of advanced HNSCC is feasible. Comprehensive elastographic and perfusion analysis describes LN more precisely than conventional US alone, because multiparametric US delivers many objectifiable values.”

Second, the authors report that the TIC analysis on the US machine took 10-15 minutes. Do the authors find the analysis feasible in a clinical diagnostic workup? It would not be in our setting, where we would prefer to take an FNA (2-5 minutes) to clarify malignancy instead. 

Answer: Thank you for this comment, Of course, the comprehensive analysis is time consuming. Therefore, we explained in Material and Methods, that we investigated clinically suspected lymph nodes before the start of chemoradiation. Our primary goal was not the differential diagnosis of benign or malignant lymph nodes. We aimed to assess baseline characteristics of clinically suspected lymph node metastases before the start of chemoradiation. We believe that our primary goal is now clearly described in the manuscript. We added: “However, detailed multiparametric analysis is time consuming and may therefore be reserved for selected cases which may profit, like patients before the start of chemoradiation”.

Personal note: Of course, the detailed TIC analysis outside a study protocol could only be performed in an interdisciplinary US center, where almost full time sonographers are doing the scan and the analysis.

Minor comments: 

You write the study is retrospective, but the patients' sign informed consent to participate in the CEUS examination. Please explain. 

Answer: In the Material and Methods section we stated that we performed the examinations as part of our routine clinical work up before the start of chemoradiation of HNSCC. All patients were informed about the “off label” use of CEUS for lymph node diagnostics of the Head and Neck. In accordance with comments of Reviewer 1 we added a new table 1 presenting an overview of our routine multiparametric Ultrasound examination protocol and additionally performed analysis according to study protocol. We completely revised the Material and Methods section.

Please specify which software you used for the TIC analysis to evaluate TTP and AUC for velocity and volume of contrast enhancement?

Answer: As stated in the material and methods section, we used the machine integrated software of Loqic E9 GE Healthcare.

Please use English titles for ultrachall publications.

Answer: We revised the reference list.

The manuscript has many technical abbreviations making it difficult for the reader to follow. Please use the full name of TIC, TTP, PI, AUC.. so it is easier to read. 

Answer: Thank you for this important comment. We reduced abbreviations as recommended throughout the manuscript.